# Environmental characterization of the estuarine zone of the Gulf of Montijo, province of Veraguas, Panama

**Ana Luisa García**[1]*, **Diana Araúz**[1], **Eridna Martínez**[2], **Jay Molino**[1,3]

1 Centro de Biotecnología, Energías Verdes y Cambio Climático, Universidad Especializada de las Américas (UDELAS), Corregimiento de Ancón, Panama City, Republic of Panama, 2 Grupo Trenza, Panamá, Republic of Panama, 3 Sistema Nacional de Investigación (SNI), SENACYT, Panama City, Republic of Panama

* ana.garcia.4@udelas.ac.pa

**Data Availability Statement:** The data is available in the following DOI. DOI: dx.doi.org/10.17504/protocols.io.6qpvr6b3bvmk/v1.

**Funding:** The authors would like to thank the Secretaría Nacional de Ciencia, Tecnología e

## Abstract

The study was performed in the Republic of Panama. Panama has a coastline of 2,988.3 km, of which 1,700.6 km corresponds to the Pacific coast. Most of the coast is flat, and several geomorphological features characterize the Panamanian coastal sector, such as the Gulf of Montijo, which is located towards the west of the Panamanian Pacific coast in the province of Veraguas. The Gulf is a remarkable ecosystem of mangroves well preserved and internationally recognized as a Ramsar area. It represents the buffer zone of Coiba Island (Coiba National Park). Sixteen sampling stations were established along the coast to analyze the vertical and spatial variability of physical and chemical parameters (temperature, salinity, dissolved oxygen, pH, and conductivity). The quality of unexposed marine sediment was evaluated in four samples from the western sector of the Gulf of Montijo. The marine sediment samples were collected with a Dietz-LaFond Snapper dredge, between 1.5 and 3.0 m depth. Physical and chemical observations and marine sediment sampling were carried out during high tide +4 to low tide +3, in negative syzygy phase tide. The analysis of the oceanographic conditions, especially the physical and chemical properties of the water along with the longitudinal profile, showed a conservative distribution increasing gradually towards the outer region in an approximately linear way due to the dilution and mixing processes, both in the surface layer and at the bottom, and a not very developed vertical gradient, with slight stratification. The results of heavy metal analyses in marine sediments report high cadmium concentrations along the west coast, with concentrations above the limit levels of the Canadian marine sediment quality guidelines. The study area requires continuous monitoring that is representative of seasonality (dry, intermediate, and rainy periods), including a more significant number of stations since it is evidence of an affectation of the environmental quality of the marine ecosystem due to possible anthropogenic activities.

Innovación ["National Secretariat of Science, Technology, and Innovation"] (SENACYT) for financing the project "Interaction between mangrove, productive activities, and natural resources in the community of La Playa, Gulf of Montijo (FID-066)", to the community of Playa El Pito for the logistical support for the execution of the field activities. The Grant MINBUZA-2020.926889 from the Department of European Integration, The Kingdom of the Netherlands also supported the researchers. The funders had no role in study design, data collection, analysis, publication decision, or manuscript preparation. None of the authors received salaries from the funders.

**Competing interests:** The authors have declared that no competing interests exist.

## Introduction

About 80% of the pollutants that affect coastal strips are transported by rivers that flow into the sea. Rivers have the particularity to concentrate the contaminants they capture in the basins of some critical points on the marine coast (Morales-Ramírez et al., 2016; Parada V. et al., 2001a). These environments support diverse communities and ecological functions of ecosystems, but the presence of contaminants in sediments can negatively impact their environmental balance and, indirectly, human health. Most species are characterized by sensitivity to pollutants, relatively broad tolerance for a wide range of salinity, sediment particle sizes, and organic matter content, which facilitates tests with various sediment types. Ecological significance (important links in coastal food chains) and wide geographic distribution of some species allow for regional comparisons [1, 2].

According to their origin, heavy metals can be of two types: geological or anthropogenic; The first refers to the presence and distribution of metals in minerals and rocks; the second arises from productive activities in the agricultural, industrial and urban sectors and generates pollutants that are deposited in rivers, sediments, and soils [3, 4]. The enrichment of heavy metals in marine sediments is a topic of great interest, given their capacity to accumulate these chemical species and the danger they represent for the health of benthic organisms.

For an in-depth analysis of the challenge, it is essential to begin by identifying the sources from which pollutants originate, the routes by which they reach the estuaries and the sea, the flows and dynamics they experience in the marine environment, and the way it affects the ecosystems and coastal inhabitants [5, 6]. Panama's population has increased by a factor of 10 during the last 80 years, especially along coastal areas. Commercial human activity has also increased; therefore, solid waste has increased in the gulfs. The aforementioned is characteristic of a developing country [7, 8].

The Gulf of Montijo is considered a marine-coastal body of water, where freshwater and seawater mixing processes occur during the rainy season and where the movement of its regions, river zone, mixing zone, and adjacent sea zone, are dynamic. Each zone's geographic positions vary continuously from temporal to geological time scales. It is a vital fishing area with 20,910 ha of mangroves, 3,330 ha of swamp forest, and 5,3490 ha of water mirrors and estuaries. It is characterized by mudflats, coastal mangrove furrows, sandy sediments, estuarine mangroves, alluvial mangroves, and swamp forests [9, 10]. Several rivers, among them, drain this marine-coastal body of water: San Pablo, Río de Jesús, San Pedro in the North, Río Suay and Ponuga to the east, and the west by the Caté, Cañazas, and San Andrés, and has a connection with the Pacific Ocean through wide channels with relative depths of no more than 20 meters. To the west is the channel between Punta Brava and Gobernadora Island, and to the east, between Cebaco Island and Punta Alto Viejo. The remaining part of the Gulf is very shallow, with sandbanks and shallows emerging at low tide [11].

There is a strong relationship between the inhabitants, their economic activities, and the environment. The thirty-six communities bordering the Gulf of Montijo are mainly fishermen and extractors of *Anadara tuberculosa*, and they report fishing a variety of marine species [12]. Among the main activities that generate impacts on environmental quality is the increase of the agricultural frontier in open fields, streams, and river headwaters, especially the inadequate management of agrochemicals, wastewater discharge, solid waste, and fuel [9]. Due to the dependence of the inhabitants on fishing and tourism activities, the analysis of the vertical and spatial variability of physical and chemical parameters such as temperature, salinity, dissolved oxygen, pH, and conductivity and their correlations is imperative. The present study focuses on studying such physical and chemical parameters and the concentrations of heavy metals in the unexposed marine sediment of the western part of the Gulf of Montijo.

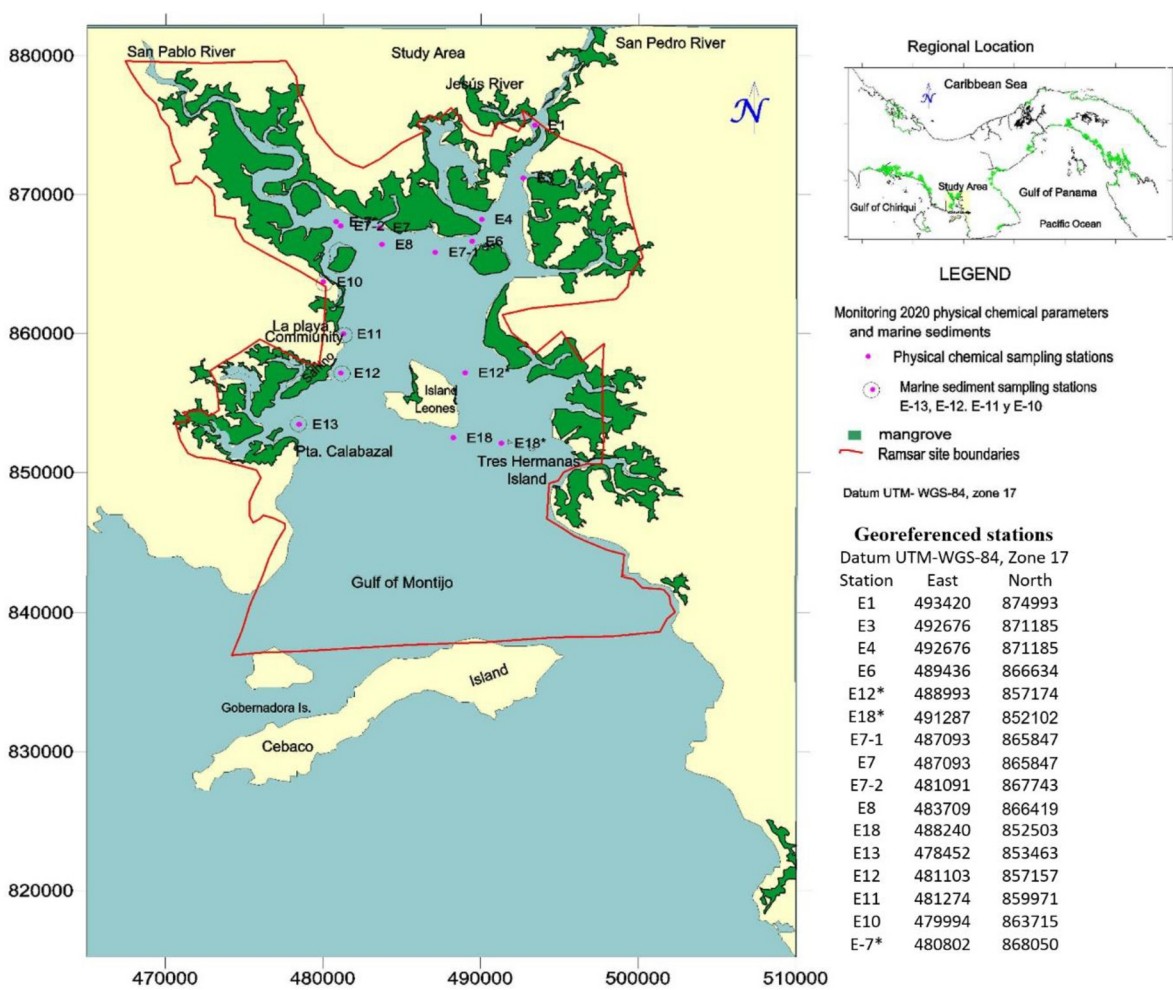

**Fig 1. Area of study and distribution of stations.** Republished from *Surfer*® *(Golden Software, LLC)* under a CC BY license, with permission from *Surfer*® *(Golden Software, LLC)*, original copyright *Surfer*® *(Golden Software, LLC)*.

## Materials and methods

### Sampling stations

The selection and number of stations (Fig 1) were based on previous studies [9]. Sixteen sampling stations were established along the coast, where physical and chemical parameters were measured from the surface to the bottom. We employed The Xylem YSI 556 MPS probe. Marine sediment was collected at four stations; measurements and sample collection were carried out during high tide +4 to low tide +3, during the negative syzygy tide phase, corresponding to October 17, 2020. The type of tide is semi-diurnal meso-tidal, greater than two meters but less than four meters; the measurements were made under the conditions described to obtain broader information on dispersion and to measure the dilution capacity and mixing processes of the water body. All stations were georeferenced with GPS-MapSource 79 Sxc global positioning in WGS 84, as shown in Fig 1.

### Physical and chemical parameters

Sixteen (16) stations were established for in situ measurements incorporated to study horizontal and spatial distribution. A profile was taken in the North-South axis to study the vertical

and longitudinal distribution of the measured physical and chemical parameters. It comprises six (6) stations from station E-1 (San Pedro River), E-3, E-4, E-6, E-12*, and E-18*; the bathymetry included in the vertical profiles is approximate. It is based on in situ measurements with a 400 kHz hand-held probe.

The Golden Surfer Software, version 7.0, was used for data analysis. We applied the Kriging interpolation method based on geostatistical or georeferenced variable theory [13, 14]. It is a method of weighted moving averages used to interpolate values from a data set obtained from a "grid" of points to obtain contours that define an area with homogeneous values of conservative parameters.

## Marine sediments

For marine sediment sampling, the physical characteristics of the seabed sediment granulometry (fine particles < 63 μm), since they concentrate the highest proportion of metals [15, 16], depth, and the possibility of penetration of the sampling equipment were taken into account. Four sediment stations were established in the western sector (E-13, E-12, E11, and E10), where in situ measurements of physical and chemical parameters were also made. Samples were extracted with the Dietz-LaFond Snapper dredge, KAHLSICO model No. 214WA130, Dietz-LaFond, between 1.5 and 3.0 m depth. The area of the study and the distribution of the stations is shown in Fig 1. Samples were stored in inert polypropylene bottles at temperatures exceeding 4 °C and analyzed with a laboratory-certified heavy metal analysis methodology; the analyses were performed in triplicate, and the results were statistically analyzed (mean, standard deviation, and variance).

The results obtained for heavy metals (As, Cd, Cu, Pb, Cr, Hg, and Ni) were compared with the Canadian Environmental Quality Guidelines (CEQG). This standard determines the ISQG (Interim Sediment Quality Guideline) values: TEL (Threshold Effect Level), a concentration below which no adverse biological effects are expected, and PEL (Probable Effect Level), a concentration above which adverse biological effects are frequently encountered (Table 1).

The degree of contamination of marine sediments by heavy metals was evaluated with the Geoaccumulation index of Müller [17–19]. This index is calculated using the following equation:

$$\mathrm{Igeo} = \log_2\left(\frac{Cn}{1.5Bn}\right)$$

Cn is the concentration of n metal in the sediment; Bn is the background metal concentration, and 1.5 is the correction factor for lithogenic effects. The background values needed to

**Table 1. Canadian sediment quality guidelines.**

| Parameters Metals | CEQG | |
|---|---|---|
| | TEL (μg/g) | PEL (μg/g) |
| Arsenic | - - - - - - | - - - - - - - |
| Cadmium | 0.70 | 4.21 |
| Copper | 18.7 | 108.30 |
| Chromium | 52.30 | 160.00 |
| Lead | 30.20 | 112.00 |
| Mercury | 0.13 | 0.70 |
| Nickel | 15.90 | 42.80 |
| Zinc | 124,00 | 271,00 |

calculate the geoaccumulation rate are taken from those found in sedimentary shales [20], and these values (in mg kg-1 dry weight) are as follows: cadmium (0.22), chromium (90), copper (45), lead (20). The extent of contamination ranges from 0 (not contaminated) to 6 (very heavily contaminated).

The degree of correlation (Pearson's linear correlation) between heavy metals and other essential parameters is often used to indicate their association level [21]. In the study of the correlation matrix, the elements (Cd and Cu) were considered with the pH physical and chemical variables.

## Results

### Vertical distribution of temperature, salinity, dissolved oxygen, and pH

**a. Temperature.** Fig 2a shows the longitudinal section of the isotherms from the surface to 20 m depth in profile 1; a quasi-homogeneous vertical temperature distribution can be observed, a water column with weak vertical gradients throughout the profile, being more evident in stations E-12* and E-18*, with values between 28.0 to 28.15 ˚C. These isotherms occupy a large area, indicating the entry of warm water into the system as a very homogeneous

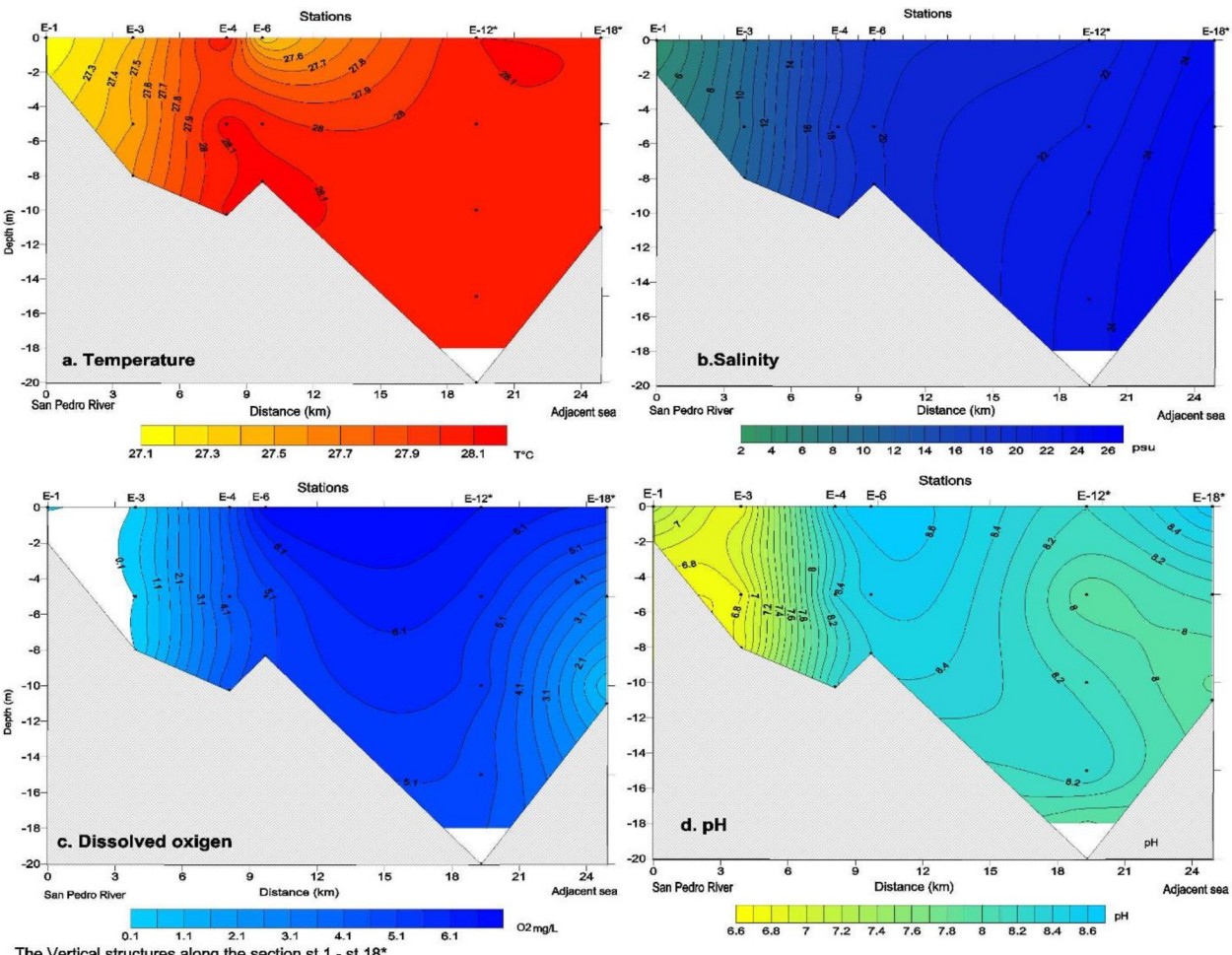

**Fig 2. Vertical distribution of temperature, salinity, dissolved oxygen and pH.** Republished from *Surfer*[R] *(Golden Software, LLC)* under a CC BY license, with permission from *Surfer*[R] *(Golden Software, LLC)*, original copyright *Surfer*[R] *(Golden Software, LLC)*.

mass of water from the bottom to the surface. Longitudinally, the water temperature gradually increases, ranging from 27.1˚ to 28.0 ˚C, from station E-1 in the inner part to station E-4. At Station E-4, a small thermal front develops due to the upward movement of the water mass from the bottom to the upper layer. Then at station E-6, a small core of slightly less warm surface water is observed, with a temperature of 27.4 ˚C, which stratifies very weakly the water column up to 5 m depth, with a vertical gradient of 0.12 ˚C/m.

The vertical temperature distribution shows that the Gulf of Montijo is a warm body of water from the surface to the bottom, quasi-homogeneous, with a very weak vertical stratification. The temperature in the longitudinal axis (profile 1) ranged from 27.11 ˚C to 28.15 ˚C. The minimum temperature was measured in the inner part at station E-1 due to freshwater discharge and the maximum at station E-18*, with an average value of 27.87 ˚C.

**b. Salinity.** The vertical distribution is presented in (Fig 2b) along a longitudinal axis from the inner part of the San Pedro River towards the outer part to station E-18* and its typical salinity/depth profile.

A gradual salinity increase from surface to bottom is observed along the section, from 3 practical salinity units (PSU) in its inner zone to more saline values in the order of 26 PSU towards the outer part at station E-18*. The typical vertical profile indicates the presence of a weak halocline at 5 m depth at all stations so that it is possible to observe a slight difference between the salinity at the bottom and the surface. The saline structure shows a horizontal gradient of 0.92 PSU/km, increasing salinity towards the outside in an approximately linear trend, both in the shallow and deep layers, and a not well-developed vertical gradient (0.13, 0.19, 0.15 psu/m, E-3, E-4, and E-12*), respectively, but visible at mid-depth. Salinity remained constant at each depth layer.

The vertical distribution indicates that the water body is slightly stratified or mixed. In this type of estuary, generally, the tidal range is large and contributes turbulent energy [22], which comes from the tidal currents within the water body. A direct way to confirm the above is through the unified salinity classification [23]. There is a ratio of surface salinity to bottom salinity (Ss/Sb) and distance normalization, the distance along the estuary divided by its length.

This classification determines that a salt wedge estuary has freshwater at the surface and oceanic water at the bottom, and a zero salinity ratio is identified. In contrast, the salinity in vertically mixed estuaries varies along the length of the estuary but is the same from the surface to the bottom everywhere. The vertically mixed estuary has a salinity ratio of one.

It was found that there are marked differences in the saline structure or vertical behavior with previous studies [9]. Those studies reported that when there is maximum rainfall (August, October, November, and December), the increase in salinity from surface to bottom is more accentuated, suggesting a highly stratified estuary condition. These differences confirm that a given estuary can exhibit stratification or mixing conditions with distance along the estuary and as a function of time covering variable scales between a tidal and seasonal cycle.

**c. Dissolved oxygen and pH.** Dissolved oxygen follows the hydrogen potential distribution pattern throughout the profile, coinciding in its minimum (0.1 mg/L at 6.6 pH units) and maximum (6.6 mg/L at 8.6 pH units) concentrations and units, respectively (Fig 2c and 2d). The pH ranged from 6.6 to 8.6, typical coastal and offshore systems values.

The water column exhibited two sections, with low dissolved oxygen concentrations between 1.0–4.6 mg/L and pH on the order of 7.0 and 8.0 at stations E-3 and E-4, respectively. Those values increase linearly towards the outside from the surface to the bottom; this structure is very similar to the salinity in that section and the second at E-18* near 8 m depth with a minimum dissolved oxygen of 1.1 mg/L and pH of 7.9.

In the inner part of the profile, from station E-1 to the limits of E-3, the hypoxic dissolved oxygen concentrations (< 1 mg/L) were measured in hypoxia condition and a pH of 6.6. Both values are related to freshwater discharge with high re-suspension and suspended solids transport, which inhibits photosynthetic processes and maximizes the activity of organic matter oxidation processes.

Optimal concentrations in the surface layer of the entire DO-pH structure are due to ocean-atmosphere exchange and photosynthetic processes, plus the dynamics of the sea, which is one of the main forcing factors affecting the distribution of oxygen and pH. These forcing factors can act independently or in combination [24].

## Horizontal distribution of temperature, salinity, dissolved oxygen, and pH

**a. Temperature.**   Less warm water isotherms on the order of 27.0 ˚C—27.5 ˚C are evident in the inner zone of the San Pedro and Jesús Rivers and towards the margins of the San Pablo River. The temperature increases slightly on the order of 0.5 ˚C and occupies much of the coastal edge of the eastern sector, extending to the south of Isla Leones in the outer part. Meanwhile, the 28.0 ˚C isotherm occupies the central part of the system. It delimits the entrance of the slightly warmer waters observed in the southwestern sector between Punta Calabazal and El Pito beach, where the highest temperatures of 30 ˚C are reached (Fig 3a).

**b. Salinity.**   Low salinities (3–10) PSU is recorded in the inner part of the Gulf due to dilution processes due to freshwater input. In the southwestern sector, salinities in the order of (26–22) PSU were obtained, maintaining a more significant effect on the adjacent seawater mass (Fig 3b). Fig 3 shows the classification diagram for the Gulf of Montijo on October 17 based on the Ss/Sb ratio. The salinity ratio is below one and above zero, indicating a slightly stratified or mixed estuary from surface to bottom.

**c. Dissolved oxygen.**   The surface distribution of dissolved oxygen (Fig 3c) follows the distribution pattern of the rest of the experimental parameters, but mainly with pH distribution which clearly defines a region of hypoxic or near anoxic oxygen concentrations < 1 mg/L in the northern and northwestern interior. Another region with low oxygen concentrations towards the southwestern sector, forming a low-level nucleus, located in the Sahino locality, maintains a radial distribution that increases from (1–5 mg/L) towards Leones Island's north, west sector towards the south up to Punta Calabazal.

**d. Hydrogen potential (pH).**   The horizontal distribution of pH is shown in Fig 3d, showing a conservative distribution between 6.8 and 8.7 pH units, from the inner part of the system towards the central part, following the salinity distribution. The increase in pH is due exclusively to increased concentrations of basic salts dissolved in seawater, which cause alkaline reactions and increase pH to typical marine values [25].

## Quality of marine sediments

The results obtained for Cd, Cu, and Pb are shown in Fig 4; the geoaccumulation index calculation values and the comparison of Cd, Cu, and Pb concentrations with Canadian environmental guidelines are shown in Fig 5. The correlation of Cu and Cd with physical and chemical parameters was analyzed using Pearson's correlation matrix, as summarized in Table 2.

**a. Cadmium.**   Cd concentrations reported values ranging from 25.979 to 73.636 mg/kg. When comparing the values obtained with the international guidelines, it can be seen that the PEL values were exceeded in all stations by 17 times the guideline value (4.2 mg/kg), a concentration above which adverse biological effects are frequently found (Fig 5).

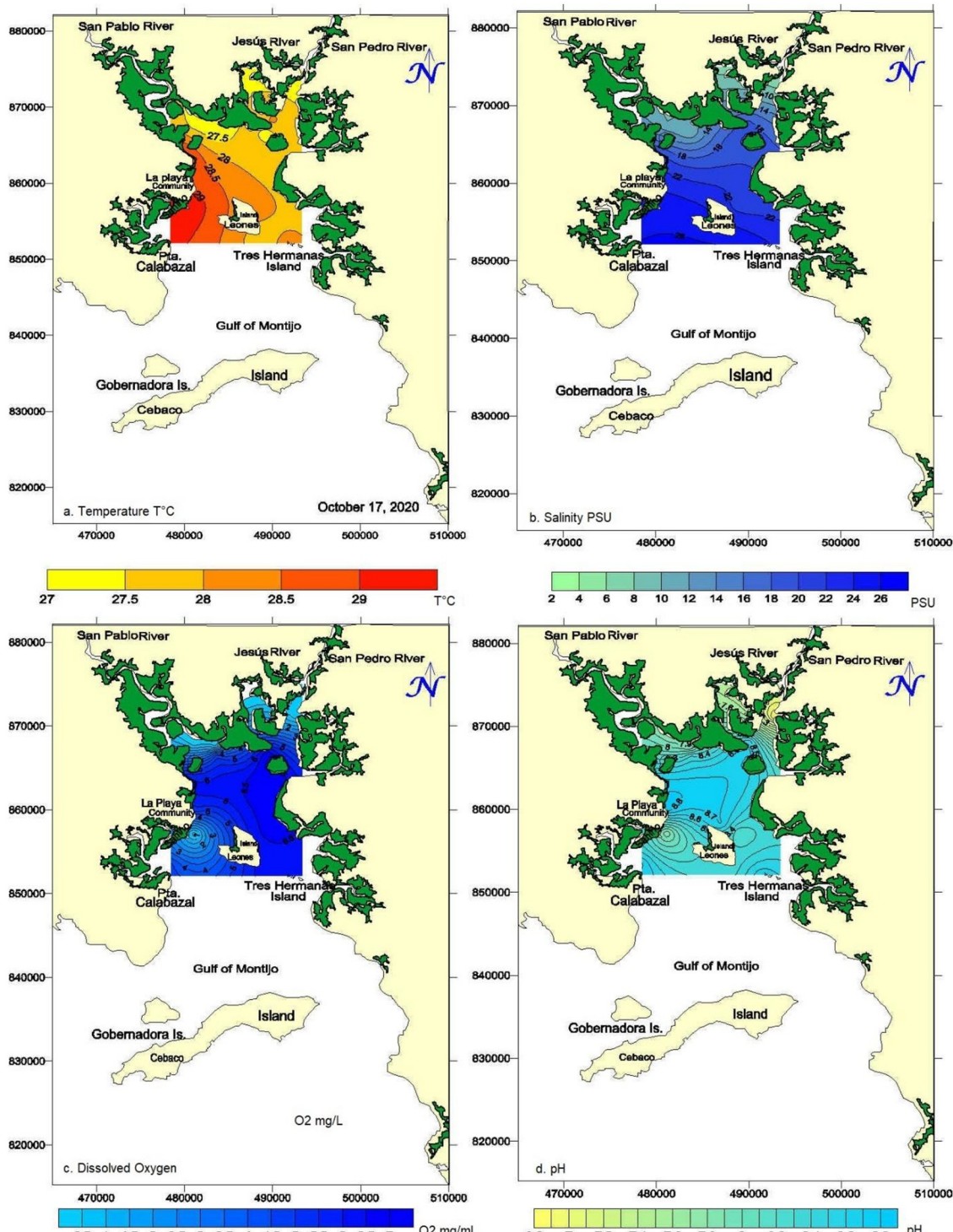

**Fig 3. Horizontal distribution of temperature, salinity, dissolved oxygen, and pH.** *Surfer*® *(Golden Software, LLC)* under a CC BY license, with permission from *Surfer*® *(Golden Software, LLC)*, original copyright *Surfer*® *(Golden Software, LLC)*.

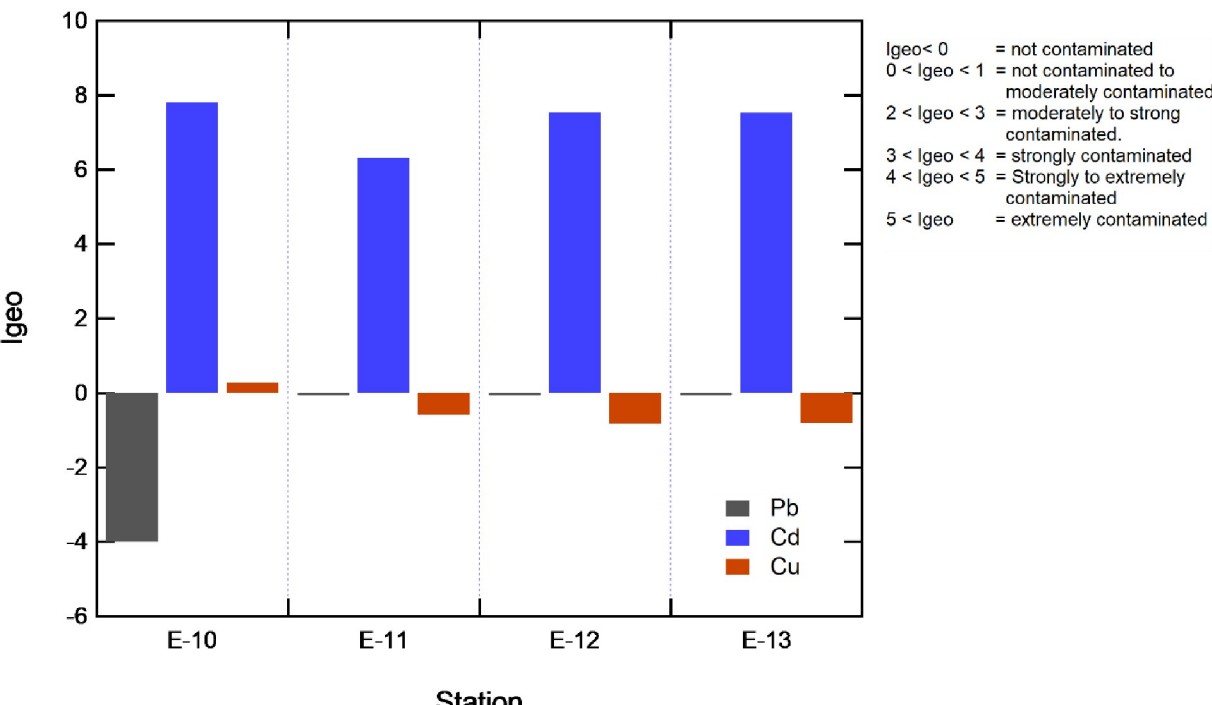

**Fig 4. Geoaccumulation indez for Cadmiun (Cd), Copper (Cu), and Lead (Pb).**

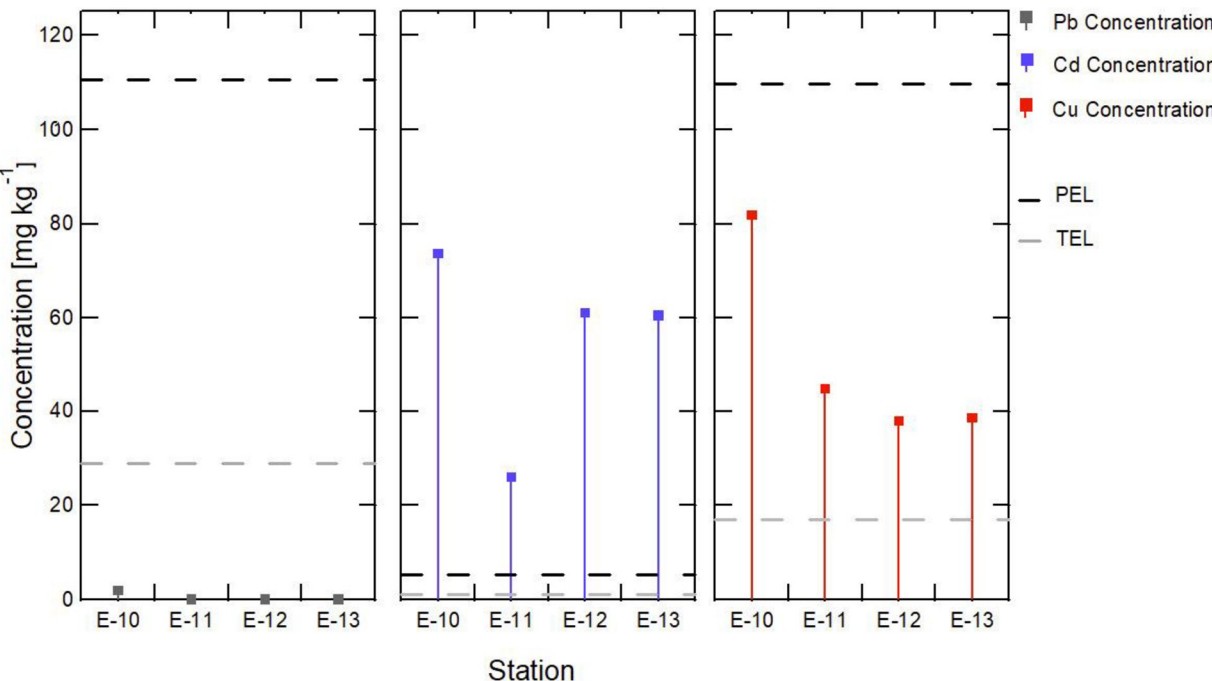

**Fig 5. Cadmium (Cd), Copper (Cu), and Lead (Pb) concentration at Stations E10-E13.**

**Table 2. Pearson correlation coefficients.**

| Parameters | Cadmium | Copper | Temperature | Salinity | pH |
|---|---|---|---|---|---|
| Cadmium | 1 | | | | |
| Copper | 0.47311944 | 1 | | | |
| Temperature | -0.61258446 | 0.40612596 | 1 | | |
| Salinity | -0.48184356 | -0.77258251 | -0.17665344 | 1 | |
| pH | 0.95404149 | 0.23873837 | -0.77981699 | -0.42080691 | 1 |

The degree of sediment contamination (Igeo) for this metal is strongly contaminated throughout the stations: E-10, E-11, E-12, and E-13, reaching a geoaccumulation index on the order of 7.96; 6.45; 7.69, and 7.68, respectively, exceeding the maximum class of 6.

**b. Copper.**   Copper concentrations range from 37.912 to 81.891 mg/kg, with station E-10 reporting the highest concentration, exceeding the PEL value (52.03 mg/kg), a concentration above which adverse biological effects are frequently found. While stations E-11, E-12, and E-13 report concentrations lower than the PEL but higher than the threshold, ISQG (TEL), concentrations below which adverse biological effects are not expected (Fig 5). The geoaccumulation index in the sediments of station E-10 classifies it as slightly contaminated, and in the remaining uncontaminated stations, the Igeo is negative. Therefore, the concentrations are lower than the reported value, or the Cu concentrations are of lithogenic origin.

**c. Lead.**   At the E-10 station, the Pb concentration was 1.867 mg/kg, this value being below the threshold, ISQG (TEL); a concentration below which no adverse biological effects are expected (Fig 5). The geoaccumulation index in the sediments of station E-10 classifies them as uncontaminated; the Igeo is negative.

## Pearson correlation matrix

The degree of correlation (Pearson's linear correlation) between heavy metals and other important parameters is often used to indicate their association level [21]. The correlation matrix considered the elements (Cd and Cu) with the pH variable measured at the bottom of the water column (Table 2). Significant positive correlations (p = 0.05) were observed between cadmium and pH (r = 0.95).

## Discussion

The saline structure shows a horizontal gradient of 0.92 PSU/km, increasing towards the outer part in an approximately linear way, both in the shallow and deep layers, and a not very developed vertical gradient (0.13, 0.19 and 0.15 psu/m, E-3, E-4, and E-12*), respectively. Both vertical and horizontal distributions of dissolved oxygen indicate areas with depletion problems of this element below 5 mg/L, and some show near anoxia, i.e., values < 1 mg/L, which could be due to eutrophication. These environments are known as large sinks with environmental stressors, such as agriculture and the discharge of untreated wastewater. Organic matter and the discharge of untreated wastewater are prevalent in this area. Therefore, the depletion of dissolved oxygen concentrations could contribute to or be a limiting factor to the relative abundance of living organisms. Oxygen concentrations below 5 mg/L can affect the functioning and survival of biological communities, and below 2 mg/L can lead to the death of most of the ichthyofauna.

The range ranged from 6.6 to 8.6 pH units, with normal coastal and marine systems values. The increase in pH is mainly due to increases in the concentrations of basic salts dissolved in

seawater. The optimal concentrations in the surface layer of the entire DO-pH structure are due to ocean-atmosphere exchange and photosynthetic processes, without leaving aside the dynamics of the sea, which is one of the main forcing factors that affect the distribution of oxygen and pH. These forcing factors can act independently or in combination [26]. The lowest pH values recorded match the decay of dissolved oxygen and low salinities in the inner part or mouth of the San Pedro River. The decomposition of organic matter from the mangrove fringe may be responsible for low pH values.

These results show differences from previous studies. In August, October, November, and December, when maximum rainfall occurs, the increase in salinity from the surface to the bottom is more accentuated, suggesting a highly stratified estuary condition [27]. During this season, the salinity structure or vertical behavior defines a well-mixed or slightly stratified estuary. These differences corroborate that a given estuary can present stratified or mixed conditions not only with distance along the estuary but also as a function of time covering short and long variable scales, such as between a tidal cycle and seasonal periods, respectively.

Both vertical and horizontal distributions of dissolved oxygen indicate that there are areas with depletion problems of this element below 5 mg/L. Some show near anoxia or values < 1 mg/L, according to dissolved oxygen concentrations, are affected by interaction with the atmosphere (in the surface layer), biological activity, and physical and chemical processes [28]. The decreases in dissolved oxygen concentrations, mainly in the inner part of the study area, may be due to the crucial contributions of suspended sediments by river discharge. These conditions were observed during the campaign.

The Cd concentrations reported for stations E-10 to E-13 in concentrations between 25.979 to 73.636 mg/kg and an accumulation index higher than six indicate that the sediments are heavily contaminated by this metal, making it an environmental problem due to its toxicity potential. This condition reflects a substantial anthropogenic influence on the marine ecosystem. Bioaccumulation processes are reported in the gonadal tissue of Anadara tuberculosa; in the Farfán estuary, Gulf of Montijo [29]. This species is commercially exploited, which could seriously threaten the ecosystem, impacting consumers' health of this type of food [30]. Contamination by trace metals is associated with a risk to human health. Organisms can bioaccumulate some metals and transfer them through the trophic chain, producing a biomagnification effect at the highest levels. The risk is established when incorporated into the human diet [31]. Cadmium is an environmental problem with health implications due to its persistence and long biological half-life (10–40 years) in the human body, especially in the kidneys [32].

The conditions existing during the sampling campaign indicate highly significant positive correlations ($p = 0.05$) between cadmium and pH ($r = 0.95$), indicative of accumulation in the marine sediments in the western sector. An increase in pH near anoxic conditions would cause the precipitation of metals as insoluble sulfides, which would immobilize and accumulate in the sediment [33].

Future work focuses on characterizing the estuarine zone during the dry season (summer) and intermediate and rainy seasons to obtain data to analyze the variations that may occur depending on the seasonal period. The most significant changes occur in the inputs of substances that reach the estuary by surface runoff.

## Conclusions

Oceanographic conditions along the longitudinal profile showed an approximately linear distribution, gradually increasing towards the outer Gulf, both in the surface layer and at the bottom, and a poorly developed vertical gradient, with a slight stratification, due to dilution and

mixing processes. Water temperature gradually increases longitudinally, ranging from 27.1 ᵒC to 28.0 ᵒC, from station E-1 in the inner part to station E-4.

Heavy metal contamination, particularly significant cadmium contamination in marine sediments, is critical to the ecosystem. Environmental stresses caused by human activities have risen, and evidence of bioaccumulation processes in marine biota can be found in the system.

Regarding vertical behavior, mainly in the saline structure, we found that the estuary is slightly stratified or mixed, which shows that an estuary can present differences at short scales, as in a tidal cycle, and at seasonal and temporal scales.

Depleting dissolved oxygen concentrations near anoxia conditions in the inner part and southwest sector of the Gulf is a critical condition for the ecosystem's health. Since there is an increase in environmental stressors, environmental authorities should be aware of these characteristics and continue with their environmental patrol to strengthen decision-making to protect this aquatic ecosystem.

## Author Contributions

**Conceptualization:** Ana Luisa García, Diana Araúz, Eridna Martínez.

**Data curation:** Ana Luisa García, Diana Araúz, Eridna Martínez, Jay Molino.

**Formal analysis:** Ana Luisa García, Diana Araúz, Eridna Martínez, Jay Molino.

**Funding acquisition:** Ana Luisa García, Diana Araúz, Eridna Martínez, Jay Molino.

**Investigation:** Ana Luisa García, Diana Araúz, Eridna Martínez.

**Methodology:** Ana Luisa García, Diana Araúz, Eridna Martínez.

**Project administration:** Ana Luisa García, Diana Araúz, Eridna Martínez.

**Resources:** Ana Luisa García, Diana Araúz, Eridna Martínez, Jay Molino.

**Software:** Ana Luisa García, Diana Araúz, Eridna Martínez.

**Supervision:** Ana Luisa García, Diana Araúz, Eridna Martínez.

**Validation:** Ana Luisa García, Diana Araúz, Eridna Martínez, Jay Molino.

**Visualization:** Ana Luisa García, Diana Araúz, Eridna Martínez, Jay Molino.

**Writing – original draft:** Ana Luisa García, Diana Araúz, Eridna Martínez, Jay Molino.

**Writing – review & editing:** Ana Luisa García, Diana Araúz, Eridna Martínez, Jay Molino.

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
