## [Decision Letter · Decision Letter 0]

29 Jun 2022

PONE-D-22-11177Environmental Characterization of the Estuarine Zone of the Gulf of Montijo, Province of Veraguas, Panama.PLOS ONE

Dear Dr. García,

Thank you for submitting your manuscript to PLOS ONE. After careful consideration, we feel that it has merit but does not fully meet PLOS ONE’s publication criteria as it currently stands. Therefore, we invite you to submit a revised version of the manuscript that fully addresses the points raised by the three reviewers during the review process. All the commentas should be considerered and changed accordingly, in order to overcome all the flaws reported.

We look forward to receiving your revised manuscript.

Kind regards,

João Miguel Dias, Ph.D.

Academic Editor

PLOS ONE

Journal Requirements:

5. Please ensure that you include a title page within your main document. You should list all authors and all affiliations as per our author instructions and clearly indicate the corresponding author.

6. We note that Figure 1 and 2 in your submission contain map images which may be copyrighted. All PLOS content is published under the Creative Commons Attribution License (CC BY 4.0), which means that the manuscript, images, and Supporting Information files will be freely available online, and any third party is permitted to access, download, copy, distribute, and use these materials in any way, even commercially, with proper attribution. For these reasons, we cannot publish previously copyrighted maps or satellite images created using proprietary data, such as Google software (Google Maps, Street View, and Earth). For more information, see our copyright guidelines: http://journals.plos.org/plosone/s/licenses-and-copyright.

 a. You may seek permission from the original copyright holder of Figure 1 and 2 to publish the content specifically under the CC BY 4.0 license. 

7. Please include a separate caption for each figure in your manuscript.

Reviewers' comments:

Reviewer's Responses to Questions

**Comments to the Author**

1. Is the manuscript technically sound, and do the data support the conclusions?

Reviewer #1: Partly

Reviewer #2: Partly

Reviewer #3: Yes

2. Has the statistical analysis been performed appropriately and rigorously? 

Reviewer #1: Yes

Reviewer #2: No

Reviewer #3: Yes

3. Have the authors made all data underlying the findings in their manuscript fully available?

Reviewer #1: Yes

Reviewer #2: No

Reviewer #3: Yes

4. Is the manuscript presented in an intelligible fashion and written in standard English?

Reviewer #1: Yes

Reviewer #2: Yes

Reviewer #3: Yes

5. Review Comments to the Author

Reviewer #1: Please see the following comments and suggested revisions which are predominantly related to grammar and organization.

Abstract

I recommend revising the introductory sentences to introduce study relevance. The current introductory sentences are taken from the beginning of the Materials and Methods section without context (e.g., “sampling stations were established along the coast...” though the study location is not introduced until the second sentence).

Introduction

Line 12: Remove period before parenthesis containing references.

Lines 33, 98: Add period after parenthesis containing references.

Materials and methods

Lines 50-54: What time of year did measurements occur? Any repeated sampling or did all measurements occur on a single day? The only date I noticed was on Line 190 (October 17).

Line 62: What type of hand-held probe was used (model, manufacturer)?

Line 64: What software/package was used for kriging?

Lines 88-89: Change comma to period for “1.5” in equation.

Results

Lines 128-134, 188, 256-258: Practical salinity units should be abbreviated “psu” instead of “ups”.

Line 140: “bottom salinity” should not be capitalized.

Line 148: Referenced as “CATHALAC 2007” elsewhere.

Lines 151-153: This brings up an important point, during what time of year were these measurements taken? Repeated measurements or was this only a snapshot in time? This should be mentioned in the Materials and Methods and referenced in the Discussion where the authors describe how environmental parameters may change with seasons and tides and the potential need for additional sampling.

Line 169, 271, 312: Should “OD-pH” be DO-pH for dissolved oxygen? If not, please define this abbreviation.

Lines 195-196, 260-261: Dissolved oxygen concentrations <1 mg/L is more accurately described as hypoxic and near anoxic.

Line 207: Arsenic (As) is not shown in Figure 4 and should be removed.

Line 216: Space needed between “it” and “can”.

Lines 239-241: Since arsenic could not be quantified, I recommend this subheading should be removed and the sentence moved to the Materials and Methods in the “Marine Sediments” paragraph.

Line 244: I do not follow how a correlation matrix between elements and the abiotic factors measured here such as pH can indicate their origin, it simply shows a level of association in this case.

Discussion

Line 255: Degree symbol after “27.1” is inconsistent with remainder of manuscript.

Lines 265-267: Requires references for these kinds of generalizations since oxygen tolerances are species-specific.

Line 286: “range ranged” should be rephrased.

Line 278: “differences with previous studies” such as? Citations needed.

Line 291: “Tuñon et al.” year needed and not included in reference list.

Line 292: “Anadara tuberculosa” should be italicized.

Line 293: “which suggests and could translate” should be rephrased, e.g., “which could translate”.

Line 299: “Vahter et al.” year needed and not included in reference list.

Conclusion

Lines 308-323: Some sentences can be combined into a paragraph so that a paragraph is not comprised of a single sentence.

Lines 308-309: Cadmium should be lowercase; also the sentence “heavy metal contamination is critical” needs additional clarification. Critical for what?

Figures

Figure captions should be more descriptive so that a reader can clearly interpret the figure without having to repeatedly go back to the manuscript text (e.g., abbreviations, include more detail, location). The current captions read more like figure titles than descriptive captions. Also, the font size is very small and low image resolution makes it difficult to read.

Figure 1: A portion of the figure legend is in Spanish (e.g., mangler, Este, Norte). The station coordinates would be better represented in a table.

Figure 3: Top axis labels should read “stations”; b) Salinity units should read PSU on scale, “vertical” misspelled; b.1) Salinity units should read PSU, “typical” misspelled, should read “typical salinity-depth profile”; c) “oxygen” misspelled, does the large white area between E1 and E3 illustrate that dissolved oxygen was not measured close to the river or was it completely anoxic? Please elaborate; e) “ratio” misspelled, ratio is described in vertical distribution section of Results (Lines 140-146) but then cited in horizontal distribution section (Lines 189-191) which is a bit confusing.

Figures 2 and 3 should be switched. The captions should remain where they are, but the current Figures 2 and 3 are illustrating horizontal and vertical profiles, respectively.

Figure 4: “index” is misspelled in figure caption. Element abbreviations should be defined in caption.

Figure 5: Element abbreviations should be noted in figure caption.

Tables

Table 1: Commas should be converted to periods to represent decimals. Was arsenic measured? If not, then it should be removed from the table.

Reviewer #2: The aim of this manuscript is an environmental characterization of the estuarine zone of the Gulf of Montijo. Despite the data presented, the authors did not reach their objective and it seems that results from two different strategies are being presented, one for water and another for sediments.

To characterize an environment, several studies must be carried out on a seasonal variation of its main physical, chemical and biological parameters, as well the local weather. An increase on the trophic state or on the pollution level in the water or sediments can also be carried out.

The following is a breakdown of some points that need to be improved:

Abstract:

What is the environmental importance of this area? Are the waters from the Atlantic Ocean or the Pacific Ocean? Which the main feature on the west coast of Panama?

The right writing is physical and/or chemical, not physicochemical.

The four sediment samples are insufficient to characterize the entire study area. Results for heavy metals are insufficient.

The authors do not conclude their work, yet.

Introduction:

It could be better; the authors just described a little of heavy metals. Will be important to explain about the regional dynamic of both, gulf and estuaries. The Gulf is a marine environment not an estuarine zone. Which is the relation between the inhabitants, economic activities, and this environment? What is the main impact on environmental quality?

Line 2: Rivers only transport solutes, they are not their origin.

Lines 29-46: Is like an area description.

Material and Methods:

Page 2

Line 50: The authors showed the results of six stations for water and four stations for sediments, but they said that established sixteen stations? I didn’t understand this information.

Line 51: that’s right: physical and chemical parameters!!!

Line 53: What kind of tides dominate this area? Is not recommendable to use data for the hide tide and low tide how the same concept. The tide amplitude during syzygy phase is very large and the physical and chemical parameters have a great variation. Is very important to make different data analysis to show both influences, by land and ocean.

Line 57: Physicochemical Parameters: what is the significance of this? The right writing is in the Line 51.

The profile is near of the southeastern margin, and I think the data collection is insufficient to make a Krigging interpolation and reach the real hydrological pattern. And there are many methods to make this kind of interpolation.

The authors said that the sediments stations corresponding to the same sampling points for water samples. The Figure 1 shows a different location between sediments and water stations. It is very confused!

I think that the sediments collection grid is poorly, because four stations in the left margin did not represent the total area, nor the continental flux.

Why the authors chose the CEQG? This guideline is appropriated for temperate areas and their work is in a tropical area.

What’s the procedure to preserve the samples to heavy metal analysis without contamination during the sampling?

This section is incomplete, because is very hard to understand how the sampling was done and why this sampling design. Heavy metals are like a postfix, and there is no information about the gulf water circulation, freshwater flow, source indication and tidal dynamics to understand why its sample design.

Results:

Table 1 is not your Result; this is for the previous chapter.

The sampling design is not adequate and will not represent the entire Gulf area.

The authors need to correct the number in Figure 2 and in Figure 3, they made a confusion.

Page 5, Lines 147 – 153: Authors need to be aware that this paragraph is not their results.

Line 157: please, don't write "unit" after “pH”.

Page 7, Line 247: positive correlation between Cadmium and pH indicates a marine source, how the authors can explain this?

Discussion:

Page 7, lines 250-254 are like Conclusion, and between lines 254-261 are like Results.

The Authors not using References to discuss their results, and they repeat the Results.

Conclusions:

Page 8

It is unclear why the authors worked with heavy metals; this makes their conclusions not enough about these parameters.

Page 9

Line 313: There are not data about photosynthetic processes, and the Authors can not use how a conclusion.

line 317: This study is about the Gulf of Montijo, it is not an estuary, and there are at least three estuaries that flow into the gulf.

Reviewer #3: This manuscript refers to the Environmental Characterization of the Estuarine Zone of the Gulf of Montijo, Province of Veraguas in Panamá”, whose area undergoes anthropogenic action. It makes an important contribution to knowledge of heavy metals in the sediment which can lead to contamination of local biota. The title is adequate and it is properly placed in the context of the study. The Introduction is well structured and the objectives are clear. The results presented are clear and the analyses carried out are consistent with the objectives. The Introduction and Discussion are well supported, with recent work.

As far as I could see the manuscript had already been denied and the authors followed the suggestions given by previous reviewer with more recent references.

In my opinion, the subject is relevant, but for the publication of the manuscript it must strictly follow the formatting rules of Plos One.

1. "References are listed at the end of the manuscript and numbered in the order that they appear in the text. In the text, cite the reference number in square brackets "

2. There are words missing letter, no space between the next word, and other small errors. E.g. line 7, 136, 158. I recommend that the text should be fully revised very carefully.

3. The legends must be redone with all the information necessary for their understanding. E.g. Figure 2. Vertical distribution of temperature (A), salinity (B), Dissolved oxigen (C) and pH (D).

4. Figures 2 and 3 do not correspond with the legends. They are exchanged.

I recommend publication after careful review, following the journal's publication guidelines.

6. PLOS authors have the option to publish the peer review history of their article (what does this mean?). If published, this will include your full peer review and any attached files.

Reviewer #1: No

Reviewer #2: No

Reviewer #3: No

---

## [Author Response · Author response to Decision Letter 0]

8 Nov 2022

Dear Editor, 

We have improves the manuscript according to the reviewers comments. We attached a word file with our response. I am also copying the whole response to this section. We are very thankful for your time. 

Response to Reviewer's Comments: Manuscript PONE-D-22-11177

Journal Requirements:

Thank you very much. We employed the PloSOne template. 

2. Thank you for stating the following financial disclosure: "The funders had no role in study design, data collection and analysis, decision to publish, or preparation of the manuscript."

Currently, please address the following queries:

 Please clarify the sources of funding (financial or material support) for your study. List the grants or organizations that supported your study, including funding received from your institution.

The following paragraphs were included in the main text:

The authors would like to thank the Secretaría Nacional de Ciencia, Tecnología e Innovación ["National Secretariat of Science, Technology, and Innovation"] (SENACYT) for financing the project "Interaction between mangrove, productive activities and natural resources in the community of La Playa, Gulf of Montijo (FID-066)", to Eridenia Martínez of Grupo Trenzas for inviting us to be part of this project, to the community of Playa El Pito for the logistical support for the execution of the field activities. The researchers were also supported by the Grant MINBUZA-2020.926889 from the Department of European Integration, The Kingdom of the Netherlands. 

The funders had no role in study design, data collection, and analysis, publication decisions, or manuscript preparation. None of the authors received salaries from the funders.

 State what role the funders took in the study. If the funders had no role in your study, please state: "The funders had no role in study design, data collection and analysis, decision to publish, or preparation of the manuscript."

The statement was included in the main text

 If any authors received a salary from any of your funders, please state which authors and which funders.

None of the authors received financial benefits from the funders. 

 If you did not receive any funding for this study, please state: "The authors received no specific funding for this work."

We received funding for this work. 

Thank you very much.

3. In your Data Availability statement, you have not specified where the minimal data set underlying the results described in your manuscript can be found. PLOS defines a study's minimal data set as the underlying data used to reach the conclusions drawn in the manuscript and any additional data required to replicate the reported study findings in their entirety. All PLOS journals require that the minimal data set be made fully available. For more information about our data policy, please see http://journals.plos.org/plosone/s/data-availability. Upon re-submitting your revised manuscript, please upload your study's minimal underlying data set as either Supporting Information files or to a stable, public repository and include the relevant URLs, DOIs, or accession numbers within your revised cover letter. For a list of acceptable repositories, please see http://journals.plos.org/plosone/s/data-availability#loc-recommended-repositories. Any potentially identifying patient information must be fully anonymized. 

The data was published in a repository. 

Important: If there are ethical or legal restrictions to sharing your data publicly, please explain these restrictions in detail. Please see our guidelines for more information on what we consider unacceptable restrictions to publicly sharing data: http://journals.plos.org/plosone/s/data-availability#loc-unacceptable-data-access-restrictions. Note that it is not acceptable for the authors to be the sole named individuals responsible for ensuring data access. We will update your Data Availability statement to reflect the information you provide in your cover letter.

Thank you very much. 

The data is available in the following DOI. 

DOI: dx.doi.org/10.17504/protocols.io.6qpvr6b3bvmk/v1 (Private link for reviewers: https://www.protocols.io/private/1CFEA664132D11EDB1220A58A9FEAC02 to be removed before publication.)

5. Please ensure that you include a title page within your main document. You should list all authors and all affiliations as per our author instructions and clearly indicate the corresponding author.

6. We note that Figure 1 and 2 in your submission contain map images which may be copyrighted. All PLOS content is published under the Creative Commons Attribution License (CC BY 4.0), which means that the manuscript, images, and Supporting Information files will be freely available online, and any third party is permitted to access, download, copy, distribute, and use these materials in any way, even commercially, with proper attribution. For these reasons, we cannot publish previously copyrighted maps or satellite images created using proprietary data, such as Google software (Google Maps, Street View, and Earth). For more information, see our copyright guidelines: http://journals.plos.org/plosone/s/licenses-and-copyright. We require you to either (1) present written permission from the copyright holder to publish these figures specifically under the CC BY 4.0 license, or (2) remove the figures from your submission:

a. You may seek permission from the original copyright holder of Figure 1 and 2 to publish the content specifically under the CC BY 4.0 license. We recommend that you contact the original copyright holder with the Content Permission Form (http://journals.plos.org/plosone/s/file?id=7c09/content-permission-form.pdf) and the following text: "I request permission for the open-access journal PLOS ONE to publish XXX under the Creative Commons Attribution License (CCAL) CC BY 4.0 (http://creativecommons.org/licenses/by/4.0/). Please be aware that this license allows unrestricted use and distribution, even commercially, by third parties. Please reply and provide explicit written permission to publish XXX under a CC BY license and complete the attached form." Please upload the completed Content Permission Form or other proof of granted permissions as an "Other" file with your submission. In the figure caption of the copyrighted figure, please include the following text: "Reprinted from [ref] under a CC BY license, with permission from [name of publisher], original copyright [original copyright year]."

Golden Surfer Software, version 7.0, free access license, operating system: Windows 7 Ultimate / Enterprise / Professional / Home Premium / Home Basic / Starter / Service Pack 1 (SP1) (32/64 bits), x86 was used. The maps were elaborated from our own data for geoestatisctical reference. The maps are not copyrighted, it is our own creation, and can be published under the Creative Commons Attribution License (CC BY 4.0). The following line was included in the main text:

The Golden Surfer software, version 7.0, was used for data analysis, applying the Kriging interpolation method, based on geostatistical or georeferenced variable theory [13,14].

b. If you are unable to obtain permission from the original copyright holder to publish these figures under the CC BY 4.0 license or if the copyright holder's requirements are incompatible with the CC BY 4.0 license, please either i) remove the figure or ii) supply a replacement figure that complies with the CC BY 4.0 license. Please check copyright information on all replacement figures and update the figure caption with source information. If applicable, please specify in the figure caption text when a figure is similar but not identical to the original image and is therefore for illustrative purposes only.

Thank you very much. This observation was addressed in the previous statement.

7. Please include a separate caption for each figure in your manuscript.

Each figure has its own caption. 

Reviewer 1:

1. Abstract: I recommend revising the introductory sentences to introduce study relevance. The current introductory sentences are taken from the beginning of the Materials and Methods section without context (e.g., "sampling stations were established along the coast..." though the study location is not introduced until the second sentence).

The abstract was throughly revised. The following line was included:

The study was performed in the Republic of Panama. Panama has a coastline length of 2,988.3 km, of which 1,700.6 km correspond to the Pacific coast. Most of the coast is flat, and several geomorphological features characterize the Panamanian coastal sector, such as the Gulf of Montijo which is located towards the west of the Panamanian Pacific coast, in the province of Veraguas. The Gulf is a remarkable ecosystem of mangroves well preserved and internationally recognized as a Ramsar area. It represents the buffer zone of Coiba Island (Coiba National Park).

2. Introduction:

 Line 12: Remove period before parenthesis containing references.

 Lines 33, 98: Add period after parenthesis containing references.

Thank you very much. The observations were corrected. 

 Materials and methods

 Lines 50-54: What time of year did measurements occur? Any repeated sampling or did all measurements occur on a single day? The only date I noticed was on Line 190 (October 17).

The following fragment was included in the main text:

Corresponding to October 17, 2020

 Line 62: What type of hand-held probe was used (model, manufacturer).

The methodology section was clarified. The following sentence was included:

Sixteen sampling stations were established along the coast, where physical and chemical parameters were measured from the surface to the bottom. We employed the Xylem YSI 556 MPS probe. 

 Line 64: What software/package was used for kriging?

Golden Surfer Software, version 7.0, free access license, operating system: Windows 7 Ultimate / Enterprise / Professional / Home Premium / Home Basic / Starter / Service Pack 1 (SP1) (32/64 bits), x86 was used. The maps were elaborated from our own data for geoestatisctical reference. The maps are not copyrighted, it is our own creation, and can be published under the Creative Commons Attribution License (CC BY 4.0). The following line was included in the main text:

The Golden Surfer software, version 7.0, was used for data analysis, applying the Kriging interpolation method, based on geostatistical or georeferenced variable theory [13,14].

 Lines 88-89: Change comma to period for "1.5" in equation.

Thank you very much. The observation was corrected. 

 Results

 Lines 128-134, 188, 256-258: Practical salinity units should be abbreviated "psu" instead of "ups".

Thank you very much. The correct abbreviation was employed (psu).

 Line 140: "bottom salinity" should not be capitalized.

Thank you very much. The correction was made. 

 Line 148: Referenced as "CATHALAC 2007" elsewhere.

Thank you very much. The correction was made

 Lines 151-153: This brings up an important point, during what time of year were these measurements taken? Repeated measurements or was this only a snapshot in time? This should be mentioned in the Materials and Methods and referenced in the Discussion where the authors describe how environmental parameters may change with seasons and tides and the potential need for additional sampling.

The measurements were taken during the rainy season, on October 17, 2020. The aforementioned was included in Materials and Methods Section. The following reference was included as well:

Vega, A.J., Robles P., Y.A., Jordán, L. & J.C., Chang. 2004. Estudio Biológico pesquero en el golfo de Montijo. Informe de Investigación. Universidad de Panamá. 171 pp.

 Line 169, 271, 312: Should "OD-pH" be DO-pH for dissolved oxygen? If not, please define this abbreviation.

Thank you very much. The correction was made. It was changed to DO-pH (Dissolved Oxygen and pH) through out the document. 

 Lines 195-196, 260-261: Dissolved oxygen concentrations <1 mg/L is more accurately described as hypoxic and near anoxic.

Thanks for the observation. We modified the whole document accordingly. 

 Line 207: Arsenic (As) is not shown in Figure 4 and should be removed.

Thanks for the observation. It was removed. 

 Line 216: Space needed between "it" and "can".

Thanks for the observation. It was corrected. 

 Lines 239-241: Since arsenic could not be quantified, I recommend this subheading should be removed and the sentence moved to the Materials and Methods in the "Marine Sediments" paragraph.

The subheading "Arsenic" was moved to materials and methods in the "Marine Sediments" paragraph.

 Line 244: I do not follow how a correlation matrix between elements and the abiotic factors measured here such as pH can indicate their origin, it simply shows a level of association in this case.

The correlation matrix in our case indicates the level of association between abiotic factors and pH. Indeed, as stated by the reviewer, since the sentence also includes other important parameters to study in order to define the origin it might distract the reader.

The following sentence was included in the main text: 

The degree of correlation (Pearson linear correlation) between heavy metals and other important parameters is often used to indicate the level of association between them (Windom et al., 1989). The correlation matrix considered the elements (Cd and Cu) with the pH variable measured at the bottom of the water column (Table 2). Highly significant positive correlations (p=0.05) were observed between Cadmium and pH (r= 0.95).

 Discussion

 Line 255: Degree symbol after "27.1" is inconsistent with remainder of manuscript.

Thanks for the observation. It was corrected. 

 Lines 265-267: Requires references for these kinds of generalizations since oxygen tolerances are species-specific.

Thanks for the observation, the reference was placed: Chapman, 1992

 Line 286: "range ranged" should be rephrased.

The sentence was improved accordingly. 

The pH ranged from 6.6 to 8.6 pH, typical coastal and offshore systems values. 

 Line 278: "differences with previous studies" such as? Citations needed.

The following reference was included:

Vega, A.J., Robles P., Y.A., Jordán, L. & J.C., Chang. 2004. Estudio Biológico pesquero en el golfo de Montijo. Informe de Investigación. Universidad de Panamá. 171 pp.

 Line 291: "Tuñon et al." year needed and not included in reference list.

The following reference was included:

Tuñón, O., Chang, J., Del Cid, A., Goti, I & Gómez, A.2020: Concentración de metales pesados (Cu y Cd), en tejido gonadal de A. tuberculosa en el estero Farfán, Golfo de Montijo. Tecnociencia, Vol. 22, N°2, 227-243.

 Line 292: "Anadara tuberculosa" should be italicized.

Thanks for the observation. It was corrected. 

 Line 293: "which suggests and could translate" should be rephrased, e.g., "which could translate".

Thanks for the observation. It was corrected. 

 Line 299: "Vahter et al." year needed and not included in reference list.

The year was included and the reference list was updated. The following reference was included:

Vahter, M., Berglung, M., Nermell, B., & Akesson, A. (1996). Bioavailability of Cadmium from Shellfish and Mixed Diet in women. Toxicol Appl Pharm, 332-341.

 Conclusion

 Lines 308-323: Some sentences can be combined into a paragraph so that a paragraph is not comprised of a single sentence.

Thanks for the observation. The document was improved accordingly. 

 Lines 308-309: Cadmium should be lowercase; also the sentence "heavy metal contamination is critical" needs additional clarification. Critical for what?

Thanks for the observation. The word was Cadmium. It was corrected. 

 Figures

 Figure captions should be more descriptive so that a reader can clearly interpret the figure without having to repeatedly go back to the manuscript text (e.g., abbreviations, include more detail, location). The current captions read more like figure titles than descriptive captions. Also, the font size is very small and low image resolution makes it difficult to read.

Thanks for the observation. The document was improved accordingly. 

 Figure 1: A portion of the figure legend is in Spanish (e.g., mangler, Este, Norte). The station coordinates would be better represented in a table.

Thanks for the observation. The document was improved accordingly. 

 Figure 3: Top axis labels should read "stations"; b) Salinity units should read PSU on scale, "vertical" misspelled; b.1) Salinity units should read PSU, "typical" misspelled, should read "typical salinity-depth profile"; c) "oxygen" misspelled.

Thanks for the observation. All the recommendations were taken into account. The document was improved accordingly. 

 Does the large white area between E1 and E3 illustrate that dissolved oxygen was not measured close to the river or was it completely anoxic? Please elaborate; 

It is an area that presents anoxia, it is reported as less than 1mg/L taking as a reference what was established by Chapman, 1992: Concentrations less than 2 mg/L can have an impact on the ichthyofauna.

 e) "ratio" misspelled, ratio is described in vertical distribution section of Results (Lines 140-146) but then cited in horizontal distribution section (Lines 189-191) which is a bit confusing.

The images and their corresponding descriptions were corrected. Figure 2 and 3 belong to the horizontal and vertical profile respectively. 

 Figures 2 and 3 should be switched. The captions should remain where they are, but the current Figures 2 and 3 are illustrating horizontal and vertical profiles, respectively.

Thank you very much. The corresponding changes were made.

Figure 4: "index" is misspelled in figure caption. Element abbreviations should be defined in caption.

Thank you very much. The corresponding changes were made.

Figure 5: Element abbreviations should be noted in figure caption.

Thank you very much. The corresponding changes were made.

 Tables

Table 1: Commas should be converted to periods to represent decimals. Was arsenic measured? If not, then it should be removed from the table.

Thank you very much. The corresponding changes were made. The word arsenic was removed from Table 1.

Reviewer #2: 

The aim of this manuscript is an environmental characterization of the estuarine zone of the Gulf of Montijo. Despite the data presented, the authors did not reach their objective and it seems that results from two different strategies are being presented, one for water and another for sediments. To characterize an environment, several studies must be carried out on a seasonal variation of its main physical, chemical and biological parameters, as well the local weather. An increase on the trophic state or on the pollution level in the water or sediments can also be carried out. The following is a breakdown of some points that need to be improved:

We than the reviewer for the comments to improve the manuscript. The physical and chemical dynamics is an important tool for the management and conservation of marine ecosystems. In our study, the behavior of the water mass and its quality are defined. Likewise we provide information about the ecosystem since the capacity for renewal and assimilation of foreign substances introduced into it depends on a series of interralated physical, chemical and geological processes. Thus, to avoid the degradation of these bodies of water, it is of fundamental importance to understand the processes that affect it and must be studied in an interdisciplinary way, which involves topics different from the classical particle pollution ( which usually focuses on findings of pollutant concentrations).

1. Abstract:

 What is the environmental importance of this area? Are the waters from the Atlantic Ocean or the Pacific Ocean? Which the main feature on the west coast of Panama?

The abstract was modified as follows:

The study was performed in the Republic of Panama. Panama has a coastline length of 2,988.3 km, of which 1,700.6 km correspond to the Pacific coast. Most of the coast is flat, and several geomorphological features characterize the Panamanian coastal sector, such as the Gulf of Montijo which is located towards the west of the Panamanian Pacific coast, in the province of Veraguas. The Gulf is a remarkable ecosystem of mangroves well preserved and internationally recognized as a Ramsar area. It represents the buffer zone of Coiba Island (Coiba National Park). 16 sampling stations were established along the coast to analyze physical and chemical parameters' vertical and spatial variability (temperature, salinity, dissolved oxygen, pH, and conductivity). The quality of unexposed marine sediment was evaluated in four samples from the western sector of the Gulf of Montijo.

 The right writing is physical and/or chemical, not physicochemical.

Thank you very much. The corresponding changes were made

 The four sediment samples are insufficient to characterize the entire study area. Results for heavy metals are insufficient. The authors do not conclude their work, yet.

The objective of the study was to analyze the vertical and spatial variability of the physicochemical parameters (temperature, salinity, dissolved oxygen, pH and conductivity) and their interrelationships. In addition, we made a preliminarily evaluation of the concentrations of heavy metals in the unexposed marine sediment of the western sector of the Gulf of Montijo. Thus, a study for the heavy metals is part of an ongoing study.

 Introduction:

 It could be better; the authors just described a little of heavy metals. Will be important to explain about the regional dynamic of both, gulf and estuaries. The Gulf is a marine environment not an estuarine zone. 

The Gulf of Montijo is considered a coastal marine water body, in which processes of mixing fresh water with sea water occur during the rainy season and where the movement of its regions; river zone, mixing zone and adjacent sea zone, are dynamic and their geographical positions vary continuously from temporal scales, to geological time scales; these characteristics define it as an estuarine system. This term is broader and also covers transition areas, directly or indirectly affected by the estuary itself, according to Kejerfve, 1984.

 Which is the relation between the inhabitants, economic activities, and this environment? What is the main impact on environmental quality?

The following lines were included in the main text:

There is an important relationship between the inhabitants, their economic activities and the environment, the thirty-six communities surrounding the Gulf of Montijo since most of them are fishermen and extractors of Anadara Tuberculosa. Furthermore, they report fishing a variety of marine species (Olmo, 2004) . Among the main activities that generate impacts on the environmental quality is the increase in the agricultural borders, both in the open field, as well as in streams and river headwaters; especially due to the inadequate management of agrochemicals, wastewater discharge, solid waste and fuel (CATHALAC, 2007).

 Line 2: Rivers only transport solutes, they are not their origin.

Thank you very much. The corresponding changes were made.

 Lines 29-46: Is like an area description.

In a previous revision of the manuscript a reviewer suggested us to include this element in the main text. We also agree that it was necessary to include these lines.

 Material and Methods:

 Line 50: The authors showed the results of six stations for water and four stations for sediments, but they said that established sixteen stations? I didn't understand this information.

The writing was improved. The following change was made

A profile was taken on the North-South axis to study the vertical and longitudinal distribution of the physicochemical parameters measured. It comprises six (6) stations from station E-1 (Río San Pedro), E-3, E-4, E-6, E-12*, and E-18*, the bathymetry included in the vertical profiles is approximate . It is based on in situ measurements with a 400 kHz handheld sonde. All the stations are incorporated for the study of the horizontal and spatial distribution. → Sixteen (16) stations were established for in situ measurements, which are incorporated for the study of horizontal and spatial distribution. A profile was taken in the North-South axis to study the vertical and longitudinal distribution of the measured physical and chemical parameters. It comprises six (6) stations from station E-1 (San Pedro River), E-3, E-4, E-6, E-12*, and E-18*, the bathymetry included in the vertical profiles is approximate. It is based on in situ measurements with a 400 kHz hand-held probe. 

 Line 51: that's right: physical and chemical parameters!!!

The main text was modified accordingly.

 Line 53: What kind of tides dominate this area? Is not recommendable to use data for the hide tide and low tide how the same concept. The tide amplitude during syzygy phase is very large and the physical and chemical parameters have a great variation. Is very important to make different data analysis to show both influences, by land and ocean.

The following was included in the mian text:

The type of tide is semi-diurnal meso-tidal, greater than two meters, but less than four meters; the measurements were made under the conditions described in order to obtain broader information on dispersion and to measure the dilution capacity and mixing processes of the water body.

 Line 57: Physicochemical Parameters: what is the significance of this? The right writing is in the Line 51.

The main text was modified accordingly.

 The profile is near of the southeastern margin, and I think the data collection is insufficient to make a Krigging interpolation and reach the real hydrological pattern. And there are many methods to make this kind of interpolation.

Thank you for your comments. The georeferenced Kriging model was applied as an analytical method to determine the spatial and vertical distribution of physical and chemical parameters, which is based on the foundations of geostatistical theory or georeferenced variables. It is a method of weighted moving averages used to interpolate values of a data set obtained from a "network" of points, in order to obtain contours that define an area with homogeneous values. The Kriging geostatistical method allows considering the values of a property throughout the space by exact interpolation according to the concept of regionalized variable from the variance parameter (Armstrong & Carignan, 1997). Kriging is an unbiased linear estimator that minimizes the variance of the estimation error. Provides a best estimate for the given variogram (graph indicating the relationship between variance and distance). In addition, Kriging allows knowing the uncertainty of the results obtained and must find the best linear estimator of this property, to preserve at the same time the minimum estimation variance. 

Armstrong M. & Carignan J. (1997). Géostatistique linéair : application au domaine minier. Paris : Les Presses of the School of Mines. 112 pages.

 The authors said that the sediments stations correspond to the same sampling points for water samples.

The following changes were made in the main text:

Four stations were established (E-13, E-12, E11 and E10), which correspond to the same sampling points for the physicochemical parameters → Four sediment stations were established in the western sector (E-13, E-12, E11 and E10), where in situ measurements of physical and chemical parameters were also made.

 The Figure 1 shows a different location between sediments and water stations. It is very confused!

I think that the sediments collection grid is poorly, because four stations in the left margin did not represent the total area, nor the continental flux.

Thank you very much for your comments. Indeed your observation is correct, however the scope of this study is to analyze the vertical and spatial variability of the physicochemical parameters (temperature, salinity, dissolved oxygen, pH and conductivity) and their interrelationships. Furthermore we made a preliminary evaluation of the the concentrations of heavy metals in the unexposed marine sediment from the western sector of the Gulf of Montijo (the sectgor with hihest anthropic pressure in the region).

 Why the authors chose the CEQG? This guideline is appropriate for temperate areas, and their work is in a tropical area.

It was used only to have a way of comparing the measurements. The contamination levels were established by calculating the accumulation index (Igeo).

 What's the procedure to preserve the samples to heavy metal analysis without contamination during the sampling?

For the collection of marine sediment samples, the National Guide for the Collection and Preservation of Samples protocol has used: waters, sediments, aquatic communities, and liquid effluents, Environmental Company of the State of São Paulo - CTESB, 2011. During the colection, it was ensured that it was minimally exposed to the air and that the container was filled up to the mouth. The lid was closed and sealed and immediately placed in refrigeration. 

Support materials were used: spatulas and inert polyethylene trays. At https://www.protocols.io/: DOI: 10.17504/protocolos.io.6qpvr6b3bvmk/v1, the sampling protocol is described in detail.

 This section is incomplete, because is very hard to understand how the sampling was done and why this sampling design. Heavy metals are like a postfix, and there is no information about the gulf water circulation, freshwater flow, source indication and tidal dynamics to understand why its sample design.

The research sought to analyze physical and chemical parameters' vertical and spatial variability (temperature, salinity, dissolved oxygen, pH, and conductivity) and their interrelationships. The concentrations of heavy metals in the unexposed marine sediment of the western sector from the Gulf of Montijo, where the area's main rivers discharge directly, were also analyzed.

 Results:

 Table 1 is not your Result; this is for the previous chapter.

Thank you for your comments. The table was relocated. 

 The sampling design is not adequate and will not represent the entire Gulf area.

Thank you very much for your comments. This observation was previously addressed. 

 The authors need to correct the number in Figure 2 and in Figure 3, they made a confusion.

The corrections were made accordingly. Thank you. 

 Page 5, Lines 147 – 153: Authors need to be aware that this paragraph is not their results.

We made the correction and placed this paragraph in the discussion section.

 Line 157: please, don't write "unit" after "pH".

The correction was made. 

 Page 7, Line 247: positive correlation between Cadmium and pH indicates a marine source, how the authors can explain this?

In our study, the correlation matrix indicates the level of association between abiotic factors and pH. Reference [21] points to other important parameters to study when it is necessary to define the origin. This could be the cause of confusion. The paragraph was edited as follows:

The degree of correlation (Pearson's linear correlation) between heavy metals and other important parameters is often used to indicate their association level [21]. The correlation matrix considered the elements (Cd and Cu) with the pH variable measured at the bottom of the water column (Table 3). Significant positive correlations (p=0.05) were observed between Cadmium and pH (r= 0.95).

 Discussion:

 Page 7, lines 250-254 are like Conclusion, and between lines 254-261 are like Results. The Authors not using References to discuss their results, and they repeat the Results.

Thank you very much. The corrections were made in the main text. We discussed the results using references. 

 Conclusions:

 Page 8 It is unclear why the authors worked with heavy metals; this makes their conclusions not enough about these parameters.

Thank for your observation. The conclusion was modifed accordingly. The following text was included:

Oceanographic conditions along the longitudinal profile showed an approximately linear distribution, gradually increasing towards the outer Gulf, both in the surface layer and at the bottom, and a poorly developed vertical gradient, with a slight stratification, due to dilution and mixing processes. Water temperature gradually increases longitudinally, with values ranging from 27.1 ºC to 28.0 ºC, from station E-1 in the inner part to station E-4.

 Page 9 Line 313: There are not data about photosynthetic processes, and the Authors can not use how a conclusion.

Indeed. Thank you. Changes were made accordingly.

 line 317: This study is about the Gulf of Montijo, it is not an estuary, and there are at least three estuaries that flow into the gulf.

Thank you. It was changed to estuarine systems.

Reviewer #3: 

This manuscript refers to the Environmental Characterization of the Estuarine Zone of the Gulf of Montijo, Province of Veraguas in Panamá", whose area undergoes anthropogenic action. It makes an important contribution to knowledge of heavy metals in the sediment which can lead to contamination of local biota. The title is adequate and it is properly placed in the context of the study. 

The Introduction is well structured and the objectives are clear. The results presented are clear and the analyses carried out are consistent with the objectives. The Introduction and Discussion are well supported, with recent work.

As far as I could see the manuscript had already been denied and the authors followed the suggestions given by previous reviewer with more recent references.

In my opinion, the subject is relevant, but for the publication of the manuscript it must strictly follow the formatting rules of Plos One.

We are very thankful for the reviewer's comments. The present is our third study in a series of water contamination and toxicity studies. Our contributions brought light to a challenge now being addressed in Panama and getting sufficient media coverage for policy making.

 "References are listed at the end of the manuscript and numbered in the order that they appear in the text. In the text, cite the reference number in square brackets "

Thank you very much for your observation. The changes were made accordingly.

 There are words missing letter, no space between the next word, and other small errors. E.g. line 7, 136, 158. I recommend that the text should be fully revised very carefully.

Thank you. We revised the document as directed. 

 The legends must be redone with all the information necessary for their understanding. E.g. Figure 2. Vertical distribution of temperature (A), salinity (B), Dissolved oxigen (C) and pH (D).

The text was revised, and the necessary corrections were made. We appreciate your observation.

 Figures 2 and 3 do not correspond with the legends. They are exchanged.

I recommend publication after careful review, following the journal's publication guidelines

The necessary corrections were made. We appreciate your observation.

---

## [Decision Letter · Decision Letter 1]

8 Dec 2022

PONE-D-22-11177R1Environmental Characterization of the Estuarine Zone of the Gulf of Montijo, Province of Veraguas, Panama.PLOS ONE

Dear Dr. García,

Thank you for submitting your manuscript to PLOS ONE. After careful consideration, we feel that it has merit but does not fully meet PLOS ONE’s publication criteria as it currently stands. Therefore, we invite you to submit a revised version of the manuscript that fully addresses all the points raised during the second round of the review process by both reviewers.

We look forward to receiving your revised manuscript.

Kind regards,

João Miguel Dias, Ph.D.

Academic Editor

PLOS ONE

Journal Requirements:

Reviewers' comments:

Reviewer's Responses to Questions

**Comments to the Author**

1. If the authors have adequately addressed your comments raised in a previous round of review and you feel that this manuscript is now acceptable for publication, you may indicate that here to bypass the “Comments to the Author” section, enter your conflict of interest statement in the “Confidential to Editor” section, and submit your "Accept" recommendation.

Reviewer #1: (No Response)

Reviewer #3: All comments have been addressed

2. Is the manuscript technically sound, and do the data support the conclusions?

Reviewer #1: Yes

Reviewer #3: Yes

3. Has the statistical analysis been performed appropriately and rigorously? 

Reviewer #1: Yes

Reviewer #3: Yes

4. Have the authors made all data underlying the findings in their manuscript fully available?

Reviewer #1: Yes

Reviewer #3: No

5. Is the manuscript presented in an intelligible fashion and written in standard English?

Reviewer #1: No

Reviewer #3: Yes

6. Review Comments to the Author

Reviewer #1: Abstract: I recommend revising the introductory sentences to introduce study relevance (not corrected appropriately from the first round of review). The current introductory sentences are not presented within the main text of the manuscript and also do not provide any study relevance, just simply state where the study was conducted which belongs in the introduction or methods.

Results:

"the hypoxic dissolved oxygen concentrations (< 1 mg/L) were measured in anoxia condition" is contradictory. In general, dissolved oxygen concentrations <1 mg/L is more accurately described as hypoxic or near anoxic. This was revised in a few instances but not throughout the entire manuscript since the first round of review.

Discussion:

"These results show differences with previous studies. In the months of August, October, November

and December...". This brings up the important point that this study was only a snapshot in time (October 17, 2020). The authors describe the need for additional seasonal sampling in the abstract, but this should be highlighted in more detail in the Discussion, especially since this is a significant pitfall for this study to accurately characterize environmental conditions for the Gulf of Montijo, which likely vary seasonally.

Figures

Figure 3: Why is each environmental parameter labeled as superficial? The figure caption should also mention that these are presumably surface levels of each parameter.

Figure 4: Caption includes Spanish.

Tables

Table 1: Including Table 1 and a table of coordinates in Figure 1 is redundant.

Table 2: Was arsenic (As) measured? If not, then it should be removed from the table. (Not corrected from the first round of review)

Reviewer #3: The authors accepted the suggestions and the manuscript can be accepted for publication. However, the references aren´t formatted according to Plos One.

7. PLOS authors have the option to publish the peer review history of their article (what does this mean?). If published, this will include your full peer review and any attached files.

Reviewer #1: No

Reviewer #3: No

---

## [Author Response · Author response to Decision Letter 1]

24 Jan 2023

Response to Reviewer's Comments:

Manuscript PONE-D-22-11177-R3

Reviewer 1:

1. Abstract: "These results show differences with previous studies. In the months of August, October, November

and December...". This brings up the important point that this study was only a snapshot in time (October 17, 2020). The authors describe the need for additional seasonal sampling in the abstract, but this should be highlighted in more detail in the discussion, especially since this is a significant pitfall for this study to accurately characterize environmental conditions for the Gulf of Montijo, which likely vary seasonally.

Thank you very much for your comments. The discussion was improved accordingly, and the sentence was included in the abstract:

"The study area requires continuous monitoring representative of seasonality (dry, intermediate, and rainy periods)."

2. Results: "the hypoxic dissolved oxygen concentrations (< 1 mg/L) were measured in anoxia condition" is contradictory. In general, dissolved oxygen concentrations <1 mg/L is more accurately described as hypoxic or near anoxic. This was revised in a few instances but not throughout the entire manuscript since the first round of review.

Thank you very much for the observation. The document was improved accordingly. In all instances, the expression "near anoxia" or "hypoxia" was employed. 

3. Discussion: "These results show differences with previous studies. In the months of August, October, November and December...". This brings up the important point that this study was only a snapshot in time (October 17, 2020). The authors describe the need for additional seasonal sampling in the abstract, but this should be highlighted in more detail in the discussion, especially since this is a significant pitfall for this study to accurately characterize environmental conditions for the Gulf of Montijo, which likely vary seasonally.

The discussion was modified to include the following paragraphs:

"The decreases in dissolved oxygen concentrations, mainly in the inner part of the study area, may be due to the crucial contributions of suspended sediments by river discharge. These conditions were observed during the campaign."

"Future work focuses on characterizing the estuarine zone during the dry season (summer) and intermediate and rainy seasons to obtain data to analyze the variations that may occur depending on the seasonal period. The most significant changes occur in the inputs of substances that reach the estuary by surface runoff."

4. Figures and tables

Figure 3: Why is each environmental parameter labeled as superficial? The figure caption should also mention that these are presumably surface levels of each parameter.

Thanks for your observations. The Word superficial was removed from Figure 3 and included in the caption. The image was modified as follows:

Figure 3: Superficial distribution of temperature, salinity, dissolved oxygen, and pH

Figure 4: Caption includes Spanish.

Thank you for the observation. The caption was revised accordingly.

Tables

Table 1: Including Table 1 and a table of coordinates in Figure 1 is redundant.

Thank you very much for your comment. Figure 1 was modified to include the data in table 1, which was removed from the main text. The image was revised as follows.

Table 2: Was arsenic (As) measured? If not, then it should be removed from the table. (Not corrected from the first round of review)

We looked for arsenic, but, fortunately, it was not present (or its concentration was extremely low) in the water. Table 1 contains this information. 

Reviewer 2:

The authors accepted the suggestions and the manuscript can be accepted for publication. However, the references aren't formatted according to Plos One.

Thank you very much for your comments. 

As for the references, we have used a version of the IEEE format (the references are numbered).

---

## [Editor Report · Decision Letter 2]

13 Mar 2023

Environmental Characterization of the Estuarine Zone of the Gulf of Montijo, Province of Veraguas, Panama.

PONE-D-22-11177R2

Dear Dr. García,

We’re pleased to inform you that your manuscript has been judged scientifically suitable for publication and will be formally accepted for publication once it meets all outstanding technical requirements.

Kind regards,

João Miguel Dias, Ph.D.

Academic Editor

PLOS ONE
---

## [Editor Report · Acceptance letter]

20 Mar 2023

PONE-D-22-11177R2 

Environmental Characterization of the Estuarine Zone of the Gulf of Montijo, Province of Veraguas, Panama. 

Dear Dr. García:

I'm pleased to inform you that your manuscript has been deemed suitable for publication in PLOS ONE. Congratulations! Your manuscript is now with our production department. 

Kind regards, 

on behalf of

Prof. João Miguel Dias 

Academic Editor

PLOS ONE